Modified MobileNetV2 transfer learning model to detect road potholes

http://orcid.org/0009-0000-5365-0152 Tanwar Neha
http://orcid.org/0000-0003-4284-7464 Turukmane Anil V. anil.turukmane@vitap.ac.in
School of Computer Science and Engineering, VIT-AP University , Amaravati, Andhra Pradesh , India
Alatas Bilal
Electronic publication date: 2025 Jan 21
Publication date: 2025
Volume: 11
Electronic Location ID: e2519
Received 2024 Apr 12; Accepted 2024 Oct 24
Copyright: © 2025 Tanwar and Turukmane
Copyright year: 2025
Copyright holder: Tanwar and Turukmane
License: This is an open access article distributed under the terms of the Creative Commons Attribution License, which permits unrestricted use, distribution, reproduction and adaptation in any medium and for any purpose provided that it is properly attributed. For attribution, the original author(s), title, publication source (PeerJ Computer Science) and either DOI or URL of the article must be cited.
License URL: https://creativecommons.org/licenses/by/4.0/

Keywords: Pavement, Pothole detection, MMNV2 Model, Deep neural network, Transfer learning, Deep learning, Classification

Funding: The authors received no funding for this work.

==============================
Road damage often includes potholes, cracks, lane degradation, and surface shading. Potholes are a common problem in pavements. Detecting them is crucial for maintaining infrastructure and ensuring public safety. A thorough assessment of pavement conditions is required before planning any preventive repairs. Herein, we report the use of transfer learning and deep learning (DL) models to preprocess digital images of pavements for better pothole detection. Fourteen models were evaluated, including MobileNet, MobileNetV2, NASNetMobile, DenseNet121, DenseNet169, InceptionV3, DenseNet201, ResNet152V2, EfficientNetB0, InceptionResNetV2, Xception, and EfficientNetV2M. The study introduces a modified MobileNetV2 (MMNV2) model designed for fast and efficient feature extraction. The MMNV2 model exhibits improved classification, detection, and prediction accuracy by adding a five-layer pre-trained network to the MobileNetV2 framework. It combines deep learning, deep neural networks (DNN), and transfer learning, which resulted in better performance compared to other models. The MMNV2 model was tested using a dataset of 5,000 pavement images. A learning rate of 0.001 was used to optimize the model. It classified images into ‘normal’ or ‘pothole’ categories with 99.95% accuracy. The model also achieved 100% recall, 99.90% precision, 99.95% F1-score, and a 0.05% error rate. The MMNV2 model uses fewer parameters while delivering better results. It offers a promising solution for real-world applications in pothole detection and pavement assessment.

Introduction

Potholes are significant road defects that pose serious risks to vehicle maintenance and traffic safety (Egaji et al., 2021; Nhat-Duc, Nguyen & Tran, 2021). Their formation is primarily due to the combined effects of water infiltration and vehicular traffic. Moisture seeps into the soil beneath the road surface, weakening the pavement structure, while vehicle loads further deteriorate sections of the road (Majidifard, Adu-Gyamfi & Buttlar, 2020). In addition to being a major inconvenience, potholes threaten road safety (Guan et al., 2021). Traditionally, potholes were detected through manual visual inspection by trained inspectors and engineers. This method is time-consuming, costly, and hazardous (Atencio et al., 2022). Modern pothole detection systems analyze video or still images captured by cameras mounted on vehicles or drones. Image processing algorithms assess the shape, size, and texture of road surfaces, enabling accurate identification of potholes. Integrating these detection systems with navigation tools and road maintenance databases can enhance road condition monitoring, optimize repair scheduling, and prioritize maintenance tasks. Additionally, these systems improve road safety by preventing injuries and accidents caused by undetected potholes (Arya et al., 2021). Recent research in this field continues to expand, with substantial literature being published in the last few years. This shift from manual to automated pothole detection not only streamlines the process but also enhances road safety and reduces long-term maintenance costs.

Pothole detection methods have advanced significantly, incorporating diverse technologies to improve accuracy and efficiency. One of the most common techniques is image processing, where road images or videos are analyzed to detect potholes. Computer vision algorithms identify features such as changes in texture, color, and depth, which are used to distinguish potholes from the surrounding road surface. This method offers a reliable, non-invasive approach to monitoring road conditions (Dhiman & Klette, 2020). Sensor-based techniques are another key method. These involve the use of various sensors, such as accelerometers, global positioning systems (GPS), and laser-based sensors. They detect changes in road surface characteristics, like vibrations or height variations, which correspond to potholes. This data helps identify, map defects on the road, and provide real-time information to maintenance teams (Cao, Liu & He, 2020). Machine learning (ML) has become increasingly important in pothole detection. Advanced methods, particularly deep learning models, can predict potholes by analyzing large datasets of labeled images or sensor data. These models are trained to recognize patterns and anomalies associated with potholes, enabling more accurate predictions and classifications (Dib, Sirlantzis & Howells, 2020). Another approach is crowdsourcing, where the public is invited to report potholes using mobile apps or other tools. These reports are then used to create maps of potholes in specific areas. This method enhances data collection by using information from a broader population, leading to more comprehensive monitoring of road conditions (Peraka & Biligiri, 2020). Vehicle-based techniques involve installing sensors on vehicles to detect potholes as they travel on roads. These systems collect data on the location and severity of potholes. They provide real-time updates to drivers and road maintenance crews. This approach offers continuous monitoring and immediate feedback (Baek & Chung, 2020).

Stereo vision is a technique that mimics human depth perception by using two cameras or sensors to capture images from slightly different angles. These two viewpoints, similar to the way our eyes work, are processed to calculate the distance and depth of objects within a scene. By comparing the differences, or "disparities," between the images, stereo vision systems can construct a 3D representation of the environment. This technology is commonly used in robotics, autonomous vehicles, and augmented reality to enable machines to perceive and navigate their surroundings with spatial awareness and depth information.

Stereo vision techniques, integrated with deep-learning models, can be employed for detecting road conditions, particularly potholes, ahead of vehicles (Chen, Yao & Gu, 2020). These approaches utilize 3D imaging, machine learning, and advanced image processing to ensure accurate and timely detection of irregularities of roads. Standard image processing techniques, including region-growing algorithms, edge detection, and threshold segmentation, are employed to analyze and identify fracture features in road surfaces (Varona, Monteserin & Teyseyre, 2020). Deep learning, machine learning methods, and convolutional neural networks (CNNs) are used to assess the performance in detecting road defects such as potholes and cracks (Du et al., 2020). Image processing and artificial intelligence (AI) significantly improve the effectiveness of pavement asset management systems (PAMS). These systems facilitate risk analysis, evaluation of pavement stability, and optimization of maintenance strategies and prioritization (Wu et al., 2020). Edge detection techniques minimize computational load by converting RGB images into grayscale, enhancing the efficiency of identifying road defects (Sattar, Li & Chapman, 2021). Location-aware convolutional neural networks (LACNNs) focus on identifying discriminative areas of the road for pothole detection using 2D vision, rather than analyzing the full road environment (Park, Tran & Lee, 2021). Furthermore, a deep learning-based method has been developed to distinguish craters from surface disturbances caused by speed bumps or driver actions. This method supports the automatic detection of various road surfaces within a crowd sensing-based application environment (Guan et al., 2021).

Similarly, the k-nearest-neighbor (KNN) method is employed to classify various types of irregular pavements, including bumps, potholes, and uneven surfaces, using a modified Gaussian background model to identify pavement anomalies (Kim et al., 2022). A new random forest-based approach has been reported to automatically detect potholes by utilizing in-built vibration sensors and GPS receivers on smartphones, enabling a road quality monitoring system (Aparna et al., 2022). To assess the severity of road hazards caused by poor conditions, a road hazard index has been created, which uses smartphone-collected data on potholes and traffic (Ma et al., 2022). A threshold-based hybrid approach integrates multiple machine learning techniques to improve the accuracy of real-time detection and classification of road surface anomalies. This approach relies on sensor data from smartphones and dynamic behavioral patterns of vehicles across varying surface conditions (Atencio et al., 2022). For crater detection, three advanced object recognition systems—YOLOv4, YOLOv4-tiny, and YOLOv5—are utilized, following the “you only look once” (YOLO) methodology. These systems are evaluated for real-time performance using deep convolutional neural networks (DCNNs) (Liu, Luo & Liu, 2022).

Efficient techniques have emerged, including automated vehicle systems (AVS), which are capable of recognizing potholes and displaying adaptive driving behaviors. Vision cameras combined with CNNs are utilized to analyze and validate the characteristics of potholes and the potential driving patterns they may induce within a controlled road environment (Qureshi et al., 2023). An automated pixel-level pavement damage model, based on a modified U-Net deep learning architecture and stereo vision, effectively distinguishes between cracks and potholes in real-world conditions (Li et al., 2024). Additionally, a system integrating vision, vibration, and 3D reconstruction technology has been developed for automatic pothole detection (Zhong et al., 2022). The thermal imaging technique, using CNN-based deep learning, has been implemented to enhance the efficiency and accuracy of pothole detection systems in road environments (García-Segura et al., 2023). The advancement of various sensor technologies has facilitated the collection of two-dimensional and three-dimensional data relevant to highways. This includes devices such as Microsoft Kinect, cameras, and laser scanners. It also encompasses techniques for segmentation, three-dimensional point cloud modeling, and innovative computer vision algorithms known as SOTA. Furthermore, road pothole recognition is increasingly performed using machine learning and deep learning methodologies (Sholevar, Golroo & Esfahani, 2022).

Herein, a modified MobileNetV2 (MNV2) transfer learning model has been reported for detecting road potholes. The approach includes data distribution strategies, as well as training and testing processes. The model is designed to classify road conditions, distinguishing between normal roads and those with potholes. Performance evaluation is conducted using a confusion matrix and various metrics to assess precision and overall effectiveness.

Experimental

Several images were examined for the classification of normal pavement and potholes. A pre-trained MNV2 model was enhanced by adding five additional layers to improve classification accuracy for identifying normal and pothole images (Anil Kumar & Bansal, 2023). The dataset consisted of a total of 681 images encompassing both categories.

Data augmentation

Various data augmentation techniques were used, including vertical flipping by 180 degrees. To evaluate the performance of the proposed MNV2 model, the original dataset of 681 images was augmented to generate a maximum of 5,000 images. The enriched dataset have been made publicly accessible on Kaggle, available at https://www.kaggle.com/datasets/neha0590/normal-pothole-dataset and the Zenodo DOI identifier was https://doi.org/10.5281/zenodo.13334878. Both the ‘normal’ and ‘pothole’ categories were represented in this dataset. To facilitate data augmentation, the dataset was divided into two distinct groups with each category containing 2,500 images (Fig. 1). One thousand images, representing 20% of the dataset, were allocated for testing, while the remaining 4,000 images, comprising 80% of the data, were utilized for training the classification model. The proposed MNV2 technique employs transfer learning using a CAFFE framework, functioning as a single-shot detector (SSD) for both detection and assessment (García-Segura et al., 2023). Table 1 presents a comprehensive overview of the dataset in tabular form. The dataset developed by “Larxel” on Kaggle consists of images captured by two distinct groups (Neha, 2024). The analysis included 5,000 images of pavements, partitioned into training and testing sets. The training set comprised 80% of the data, while the testing set contains the remaining 20%.

Figure 1 Data visualization.

Table 1 Comparison of MMNV2 model parameters with different MobileNet architecture.

S. No.	Model name	Total parameters	Trained parameters	Non-trainned parameters	Total run time (S)	Accuracy (%)	
1	MobileNet	3,229,378	3,207,490	21,888	41,018	78.69	
2	MobileNetV2	2,240,986	2,206,874	34,112	35,212	81.26	
3	MMNV2 (Proposed)	2,586,434	328,450	2,257,984	7,999	99.95	

MobileNet architecture is well-suited for low-power, low-latency embedded applications. Google’s introduction of image segmentation, object identification, and image classification in 2017 has significantly advanced computer vision applications. A distinctive feature of MobileNet is its depth-wise separable convolution operation, which reduces computational requirements and parameter counts without sacrificing accuracy. This operation employs pointwise convolution to combine output channels using a 1 × 1 filter after performing depth-wise convolution with a single filter applied to each input channel. In terms of accuracy, MobileNet outperforms standard CNNs while requiring fewer computations and parameters. Several versions are available, including MobileNetV1 (Achirei et al., 2021) and MobileNetV2 (Camilleri & Gatt, 2020), which introduces enhancements such as squeeze-and-excitation modules, linear bottlenecks, and inverted residual blocks, thereby improving the accuracy and performance of the original MobileNetV1 architecture. The MobileNetV2 (MNV2) classifier exhibits superior performance compared to MobileNetV1 (MNV1) due to its incorporation of inverse residual blocks with linear bottlenecks and fewer parameters.

MobileNetV1

MobileNetV1, introduced by Gao et al. (2023) aimed to develop an efficient and lightweight neural network framework suitable for devices with limited processing power. The architecture achieves this by combining pointwise and depth-wise separable convolutions. Depth-wise convolution decomposes standard convolution into two stages: depth-wise convolution and pointwise convolution. This methodology reduces the computational cost of convolutional operations while maintaining model accuracy. Pointwise convolutions, using a 1 × 1 filter, adjust the number of feature channels in a feature map. The depth multiplier feature allows users to select the number of channels in each layer, balancing model accuracy and size. This approach has proven invaluable across various vision-related applications.

MobileNetV2

MobileNetV2 builds upon MobileNetV1, aiming to enhance the efficiency and effectiveness of neural networks. Key innovations include the use of inverted residual blocks, which consist of a lightweight bottleneck block, a linear layer, and a shortcut connection that bypasses the block. Initially, a 1 × 1 pointwise convolution reduces the number of input channels, followed by a depth-wise separable convolution that increases the number of output channels. Another 1 × 1 pointwise convolution is then applied. The term “inverted” describes this architecture, as it adds more channels before reducing them, contrasting with traditional residual blocks (Babbar & Bedi, 2023). This design improves performance while maintaining comparable model sizes, allowing MobileNetV2 to utilize greater non-linearity at a lower computational cost. Additional optimizations in MobileNetV2 include linear bottlenecks that enhance the efficiency of convolutional layers and simplify feature maps (Baduge et al., 2023). Furthermore, a specialized batch normalization technique and a feature map reduction block are incorporated to decrease feature map resolution and improve training stability. Due to these enhancements, MobileNetV2 is better suited for mobile vision applications that require real-time performance and minimal battery consumption. Table 1 compares the parameters of different MNV2 models with various MobileNet architectures. In the proposed MNV2 model, an accuracy of 99.95% was achieved, significantly surpassing the lower accuracy levels of 78.69% for MobileNetV1 and 81.26% for MobileNetV2.

Training and testing data distribution

Figure 1 illustrates that the decision-making predictive capability of the proposed MNV2 model improves with training on 4,000 images out of a total of 5,000 images. Specifically, 80% of the images are allocated for training, focusing on two classes (Liu, Liu & Wang, 2022). This strategy enhances the ability of the proposed model to identify hidden patterns and interactions due to the substantial volume of data. The extensive training dataset significantly boosts the capacity of the model to apply learned knowledge to new inputs, thereby improving its generalization ability. A larger training dataset typically enhances the accuracy and consistency of predictions of the models by providing a comprehensive understanding of all characteristics and patterns present in the data.

As mentioned, 1,000 images from the test dataset were used to evaluate the MNV2 model on new data, representing 20% of the total images analyzed. The testing dataset reflects the performance of the model in real-world scenarios. This assesses the effectiveness of the model with unseen data and identifies any overfitting or underfitting issues encountered during training. The test sample could also be used to enhance the speed and adaptability, enabling the development of models that perform well across real-world scenarios. The performance review includes the predictions made by the MNV2 model based on images from this previously unknown dataset, as depicted in the output results.

Experimental design of the proposed model

The proposed MNV2 method was developed using a deep neural network architecture in conjunction with the TensorFlow, Keras, and OpenCV libraries to classify and detect pavement images. The model incorporates distinct files containing the weights of all text (TXT) files (deploy.protext, res10 300 × 300 ssd, iter 140000.caffemodel). The CAFFE model utilizes these weights for each individual layer, alongside the files (prototxt and pbtxt) that delineate the structure of the layers. CAFFE models serve as end-to-end machine learning engines.

Experiments for this study were conducted using Jupyter Notebook, a web-based interactive computing platform that integrates live code, equations, narrative text, and visualizations. Jupyter Notebook facilitates the training of models utilizing deep learning techniques. Installed Python 3.9 libraries, including scikit-learn, pip, and pandas, were utilized for research experiments. Robust data processing capabilities of Pandas were employed to import and refresh the dataset. The efficacy of the proposed method was evaluated using data obtained from the Scikit-learn open-source library for machine learning and deep learning models. The computer system used for this inquiry was equipped with an Intel(R) Xeon(R) W-2125 processor and 64 gigabytes of random-access memory (RAM) operating at a clock speed of 4.00 gigahertz.

Methodology

The standard MNV2 architecture was extended by adding five additional layers, resulting in the proposed model. A transfer learning approach was employed by freezing all layers of the original MNV2 model to enhance feature extraction.

Modified MobileNetV2

Transfer learning allows a model trained on one task to be adapted for a similar task using a smaller dataset. Traditional machine learning often requires a large volume of labeled data to train models for specific tasks. Transfer learning, however, uses pre-trained deep learning models that can apply learned information to new tasks. This approach is particularly effective for training high-quality models when the available labeled data is limited.

(i) Initial training: A large dataset was first used to train the conventional MobileNetV2 and CAFFE models, which were then set aside for potential pre-training. These models, having been trained previously, are capable of distinguishing between normal road surfaces and potholes by recognizing patterns and general features in the data.

(ii) Refining for specific task: The pre-trained model was refined using a dataset of 5,000 images to adapt it for the specific task of pothole detection. This fine-tuning improved the focus of the model on the target task, allowing for better performance on the new dataset, which has different distribution characteristics.

(iii) Target domain evaluation: The model was evaluated on a dataset of 10,000 pavement images, despite the domain distribution differing from the training dataset. Although the proposed model performed well, the target domain had fewer labeled examples. By optimizing the performance of the model on this dataset, it was aimed to surpass its performance on the source domain. A detailed description of the architecture is provided below, along with a condensed version of Fig. 2.

Figure 2 Modified MNV2 framework.

Architecture and layer modifications

The modified MNV2 architecture benefits from the addition of customized layers, as shown in Table 2. The integration with CAFFE enhances the performance of the model in both detection and classification tasks. The MobileNetV2 structure, pre-trained on ImageNet, was imported into Keras as the foundation of the enhanced architecture. Key architectural features are shown in Table 2 and Fig. 2, which include the following:

Table 2 Additional five layers of the proposed model.

Layer name	Output size	Parameter used	
Input layer	224,224,3	0	
MobileNetV2 (frozen layers)	7,7.1280	2257,984	
AveragePooling2D	1,1,1280	0	
Flatten	1280	0	
Dense	256	327936	
Dropout	0.5	0	
Dense1	(2)	514	
Trainable parameter	328,450		
Non-trainable parameter	2,257,984		
Total parameter	2,586,434		

(i) Input images: RGB images with a resolution of 224 × 224 pixels are used as input.

(ii) Frozen MobileNetV2 layers: These layers, optimized for image classification, are frozen, meaning their weights remain unchanged during training. They capture broad visual features such as edges, textures, and shapes.

(iii) Average Pooling2D: This layer reduces the spatial dimensions of feature maps, producing a fixed-size feature vector by averaging the values across each feature map. (a) Function: The whole dataset should be subjected to global normalization. At the end of each iteration, the average and standard deviation are determined.

(b) Role: The capabilities of the model are enhanced to detect broad patterns and trends in the data. When dealing with unique datasets that display different properties, this capability becomes even more useful.

(iv) Flatten layer: The flatten layer transforms the multi-dimensional output of the previous layers into a one-dimensional array, making it suitable for fully connected layers. (a) Function: Transferring from a pooling or convolutional layer to the flattening layer is essential for a neural network’s fully connected layer to function. This adjustment is necessary for the fully linked layers that follow the flattened layer since they can only accept input in one dimension.

(b) Role: Neural networks’ flattened layer manages data flow and gets the information ready for layers that are completely linked. Using the results from the layers before creates a linear format. Given that all neurons in a completely connected layer need one-dimensional input, this translation is necessary. The neural network’s information flow is guaranteed by the flattened layer.

(v) Dense layer (256 neurons): This layer captures complex patterns from the feature vectors. It consists of 256 neurons, each connected to the previous layer’s neurons. (a) Function: There are 256 neurons in this layer, and they are all related to each other and to the neurons in the layer above them.

(b) Role: Through its role as a feature extractor, it acquires the capacity to acquire intricate hierarchical characteristics from the gathered data. The vast number of neurons that the model contains helps it comprehend the intricate correlations that are present in the data.

(vi) Dropout layer (0.5): A dropout layer with a rate of 0.5 is used to prevent overfitting by randomly deactivating 50% of the neurons during training. (a) Function: Facilitating the development of robust attributes is advised as a preventive measure against overfitting. The designated dropout rate causes the inactivation of a random 50% of the neurons in this layer during each training iteration.

(b) Role: Its regularized properties make it possible for the network to learn duplicate representations while simultaneously decreasing its reliance on any one neuron. Because of this, the model is able to generalize to data sets that were not there before.

(vii) Output dense layer (two neurons): The final layer generates class predictions for the binary classification task (normal road vs. pothole). A SoftMax activation function is used to convert the outputs into probability distributions. (a) Function: One neuron is allocated to each classification task type, constituting the final fully connected layer.

(b) Role: It is possible to get probabilities for each category from the model’s output in order to facilitate the categorization of input instances. Most of the time, the SoftMax triggering tool is used to convert raw outputs into probability distributions.

To enhance the ability of the model to extract features and classify normal and pothole instances, we added five additional layers: Average Pooling, a dense layer with 256 units, a final dense layer with a dropout rate of 0.5, a dropout layer with 1 unit, and a second dense layer. These layers were selected strategically to introduce nonlinearity, mitigate overfitting, and uncover hidden patterns in the data.

The modified MNV2 model was subjected to rigorous testing to provide insights into this design choice by comparing its performance with various layer configurations. Despite the complexity of the dataset, which includes diverse road surface conditions, the addition of these layers consistently improved the precision, convergence speed, and adaptability of the model. A rectified linear unit (ReLU) activation function was used after each convolution layer to introduce non-linear behavior. The ReLU function outputs the positive part of the convolution result, while negative outputs are set to zero. Once trained, the model was implemented for classification and detection, as shown in Fig. 3 of the block diagram. The block diagram for the proposed MMNV2 approach is described in eight phases:

Phase 1: A database containing images of both normal and pothole-affected road surfaces was used.

Phase 2: A binary classification approach was applied, with images grouped into two categories: pothole and normal. The data was then organized into two distinct folders based on these classifications. After preprocessing, the images were labeled using a graphical tool called “labeling”.

Phase 3: The combined dataset consists of 5,000 images, split between training and testing sets. Stratification was applied to ensure a balanced ratio of normal to pothole images.

Phase 4: The images were divided into two sets: the training dataset, comprising 80% of the images (4,000), and the testing dataset, comprising 20% (1,000).

Phase 5: In this phase, data exploration and preparation were performed using various batches of training and testing data before modifying the model.

Phase 6: As discussed in “Methodology”, the model was trained on the input data, and the processed outputs were evaluated against the test results.

Phase 7: To prepare the MMNV2 model for testing, the CAFFE framework and the modified MobileNetV2 were loaded.

Phase 8: Finally, the model’s predictions were validated using two sets of pavement images.

Figure 3 Process flow diagram of proposed MMNV2 model.

Image credits: data processing free icon (Iconjam; Flaticon License: https://www.flaticon.com/free-icon/data-processing_8346140); Split, table, tables icon (Axialis; Basic License: https://www.iconfinder.com/icons/2306058/split_table_tables_two_icon); Server free icon (Dreamstale; Flaticon License: https://www.flaticon.com/free-icon/server_699775); Layers free icon (bukeicon; Flaticon license: https://www.flaticon.com/free-icon/layers_3171685); Machine Learning icon (Adioma; Free license: https://adioma.com/icons/machine-learning); Settings free icon (Freepik; Flaticon License: https://www.flaticon.com/free-icon/settings_1025129); Data Processing free icons (Eucalyp; Flaticon License: https://www.flaticon.com/free-icon/data-processing_1878420?related_id=1878360&origin=search and https://www.flaticon.com/free-icon/data-processing_2857376).

Input data

The dataset used in this study was sourced from “Larxel” and is available on the Kaggle platform (Neha, 2024). It contains images of pavement, categorized into two classes: normal and pothole. As outlined in “Limitations of the Study”, this research defines two distinct categories: normal and pothole. To begin, we acquired a unique dataset comprising pavement images, which served as the foundation for our investigation.

Data pre-processing

After acquiring the images, they were organized into specific compartments to facilitate their retrieval in later stages. The data pre-processing method was employed to scan all files in the designated directories, ensuring the images were resized appropriately to meet the requirements of the proposed model. The images were then imported and converted into a dataset, with each image undergoing preprocessing to include additional metadata. Numerical data processing was handled efficiently using the Numerical Python (NumPy) library. To prepare the images for binary classification, a LabelBinarizer (LB) was used, categorizing them into two groups: 0 for normal and 1 for pothole. Following this step, the images were split into training and testing sets.

Data splitting

The dataset of 5,000 pavement images was divided into two groups: one for training and one for testing. An 80/20 split was applied, with 80% of the data (4,000 images) allocated for training and the remaining 20% (1,000 images) for testing. The training set was used to train the model and enable it to detect patterns and relationships within the dataset. The testing set, on the other hand, was used to assess the model’s performance and its ability to generalize to new data. This 80/20 division ensures the model receives sufficient data for training while allowing for a detailed evaluation of its performance.

Training the model

Figure 2 illustrates the proposed approach for designing the model. The pre-trained MobileNetV2 model was obtained using ImageNet and Keras. Subsequently, five additional layers were added to the base model, following a transfer learning approach. These layers are detailed in Table. To detect key features for classifying road conditions, extra trainable layers from the CAFFE framework were incorporated. After loading the CAFFE framework, along with the necessary TXT files and weight files, the model’s architecture was modified. This model leveraged pre-trained bias weights, reducing computational overhead without erasing learned features. Over the course of 60 epochs, with a batch size of 32, the modified MobileNetV2+CAFFE model (MMNV2) was trained. The learning rate was set to 0.001, and the Adam optimizer was used to adjust the learning rate for each weight during the training process. Transfer learning allowed for efficient updates to the neural network weights. Using pavement images, DNN model was trained and updated to make predictions.

The proposed MMNV2 model

The MMNV2 model was designed for the classification, detection, and prediction of normal and pothole conditions. It combines the CAFFE framework with a modified MobileNetV2 model. This updated version of the MMNV1 model uses CAFFE-trained MobileNetV2 to tackle issues related to detection, prediction, and classification. Data is structured as blobs, which are multidimensional arrays, to prepare images for deep learning frameworks like CAFFE and OpenCV. These blobs standardize data before inputting it into the neural network. OpenCV, a powerful computer vision toolkit, was used to process the images and detect features in both still images and video frames. OpenCV extracted regions of interest (ROI) from frames, such as pavement areas. Prior to passing the images to the neural network, they were resized, normalized, and enhanced for color. The deep neural network was then trained to differentiate between normal and pothole road surfaces. The blob function required attributes, their locations, and predictions to function effectively.

Testing the model

The MMNV2 model was used to classify and detect images of two types of pavement: normal and pothole. After loading the images, trial images were input for testing. Image elements were identified by their X and Y coordinates, and features with more than 50% confidence were located for labeling and bounding boxes. The model’s output consisted of predictions for both normal and pothole images. The testing procedure, evaluation metrics, and results are outlined below: Dataset collection: The dataset included diverse pavement images with varying backgrounds and lighting conditions, representing real-life scenarios to test the model’s robustness.

Testing procedure: The model was evaluated using static images of pavement under various conditions.

(i) Evaluation metrics: The model’s performance was assessed using metrics such as memory usage, precision, F1-score, and computation time. These metrics helped gauge the model’s accuracy and efficiency in detecting images.

(ii) Confusion matrix analysis: A confusion matrix was generated to summarize the model's performance on test data, displaying the number of correct and incorrect classifications. This analysis helped identify areas where the model struggled with pothole detection.

(iii) Comparative analysis: The MMNV2 model was compared against existing methodologies and baseline models. Various parameters, including accuracy, were used to evaluate its performance, and Fig. 4 illustrates the performance across different learning rates.

Figure 4 (A–E) The predicted tasks with different learning rates.

(iv) Results: The MMNV2 model performed well across all 13 testing scenarios, consistently providing accurate predictions. Confusion matrices provided valuable insights into areas where the model could improve, as shown in Fig. 5. Table 3 presents detailed numerical results, including memory usage, accuracy, and F1-score.

Figure 5 Performance analysis of MMNV2 model implemented with different learning rates.

Table 3 Performance evaluation of the suggested model using various learning rates.

S. No.	Learning rate	Accuracy (%)	Precision (%)	Recall (%)	F-1 score (%)	Error rate (%)	
1	0.1	97.80	95.70	100	97.8	2.2	
2	0.01	98.30	96.70	100	98.3	1.7	
3	0.001	99.10	98.20	100	99.09	0.9	
4	0.0001	98.6	97.20	100	98.5	1.4	
5	0.00001	97.20	94.60	100	97.22	2.8	

Result and discussion

The dataset consists of two classes: normal and pothole. A total of 5,000 images were used, with 4,000 images (80%) allocated for training and 1,000 images (20%) for testing (Matarneh et al., 2024; Cano-Ortiz et al., 2024). The proposed MMNV2 model achieved a remarkable accuracy rate of 99.95% in detecting and predicting both normal and pothole conditions. A confusion matrix, also known as a matrix of perplexity (Zhong et al., 2023; Daneshvari et al., 2023), is a crucial tool in statistical analysis, deep learning, and machine learning for evaluating the performance of classification models. It uses four key metrics: false positive (F+ve), false negative (F−ve), true positive (T+ve), and true negative (T−ve). These outputs illustrate the relationship between the actual and predicted classes of the dataset, providing insights into the model’s effectiveness.

Accuracy: The accuracy of a classification model is a key measure of its performance, representing the proportion of instances correctly classified by the model. Equation (1) demonstrates the equation to calculate the model’s accuracy.

(1) Accuracy=T+ve+T−veT+ve+F+ve+F−ve+T−ve=99.95%

The model's accuracy in predicting the positive class is represented by the notation T+ve. The number of times the model correctly predicts negative classes is also denoted as T−ve. Conversely, the erroneous positive class predictions made by the model are indicated by F+ve, while the erroneous negative class predictions are represented by F−ve. The compartments within the confusion matrix provide a count of instances that were accurately or inaccurately identified for each class.

Precision: Precision refers to the proportion of positive predictions that are correct, relative to the total number of predicted positive instances. The model’s precision can be calculated using Eq. (2), which measures how accurately the classification model identifies positive cases.

(2) Precision=T+veT+ve+F+ve=99.90%.

Recall: Recall measures the situation where the cost of false negatives exceeds the cost associated with the positive class. It quantifies a classification model’s ability to accurately identify instances of a specific class among all instances. Equation (3) can be used to calculate recall, also known as sensitivity.

(3) Recall=T+veT+ve+F−ve=100%.

F-1 Score: The F1-score is a metric that combines both precision and recall to evaluate the performance of a classification model. It provides a balance between these two metrics, ensuring that neither is disproportionately emphasized. The F1-score, which is crucial for assessing the effectiveness of a classification model, is calculated using Eq. (4).

(4) F1−score=(2×Precision×Recall)(Precision+Recall)=99.95%.

Error rate: The error rate is defined as the proportion of all misclassified predictions relative to the total number of predictions made by the model. It provides a measure of how often the model fails to correctly classify instances. The error rate is calculated using Eq. (5).

(5) Errorrate=F+ve+F−veT+ve+F+ve+F−ve+T−ve=0.05%.

The confusion matrix is a tool used to evaluate the performance of a classification model comprising two classes. In this matrix, the current class is arranged horizontally, while the anticipated class is organized vertically. The matrix cells contain numerical values representing the occurrence counts of the actual and predicted class combinations. Incorrect predictions are located in the off-diagonal rows and columns.

In the current customized MobileNetV2 model, which utilizes standard MobileNetV2 layers for feature extraction, there were a total of 2,586,434 parameters. Among these, 328,450 parameters were trainable, while 2,257,984 remained non-trainable, resulting in reduced computational time. In contrast, the standard MobileNetV2 model consisted of 3,540,986 parameters, which included 34,112 non-trainable and 3,506,874 trainable parameters, requiring more computational resources, as shown in Table 2. The findings indicate that at a learning rate of 0.001, the proposed MMNV2 model outperformed the other four learning rates. Higher learning rates of 0.1 and 0.01 resulted in insufficient training and erratic training patterns, leading to poor classification outcomes with misclassifications of 13 and four potholes, respectively. Similarly, training with lower learning rates of 0.0001 and 0.00001 also compromised performance, resulting in misclassifications of three and 10 potholes, respectively.

Accuracy, precision, recall, and F1-score were evaluated using the confusion matrix in the context of classification models. The diagonal cells of the matrix contain the accurate predictions, highlighting potential areas for improvement in the classification process and identifying the specific categories that contribute to misclassification. The confusion matrix also reveals any discernible patterns of uncertainty, providing a comprehensive assessment of the model's performance for each class. The matrix data indicates the following: if “X” represents both the anticipated and actual class, it signifies a true positive; if the actual class is “X” and the anticipated class is “Y,” it represents a true negative. A false positive occurs when the actual class is “Y” but the anticipated class is “X.” Conversely, a false negative arises when the anticipated class is “X,” while the actual class is “Y.” A thorough analysis of the classification model's performance was conducted using these confusion matrices. Figures 4A–4E displayed below illustrate the predicted outcomes with varying learning rates. The proposed work on the modified MobileNetV2 has been implemented with different learning rates, as summarized in Table 3.

Performance comparison of 14 different models

Table 4 presents a comprehensive comparison demonstrating that the proposed MMNV2 model significantly outperforms previous models in the detection and classification of both normal and pothole conditions. This model has demonstrated enhanced efficiency in accurately detecting and evaluating occurrences within these two classes. Given these improved outcomes, the MMNV2 model effectively addresses the complex tasks associated with pothole detection and classification. Table 5 outlines the substantial contributions and advantages of the proposed model. The findings regarding error rates, F1-scores, and performance metrics such as accuracy, precision, and recall for the testing data across fourteen distinct models—including MobileNet, NASNetMobile, DenseNet121, DenseNet169, InceptionV3, DenseNet201, ResNet152V2, EfficientNetB0, InceptionResNetV2, Xception, EfficientNetV2M, and our proposed MMNV2—are summarized in Table 4. All fourteen models were subjected to identical variable settings, including learning rate, batch size, number of epochs, and optimizer, as applied in the proposed MMNV2 model. Notably, the MMNV2 model significantly outperformed other models across various performance metrics, including total computation time. Table 5 provides detailed runtime performance measures alongside the trained and untrained parameters for all models. Figures 5A–5C illustrate the performance of different learning rates in terms of accuracy, precision, memory usage, and F1 score. Additionally, Figs. 6A–6l depict the number of instances classified correctly or incorrectly for each class.

Table 4 A comparison of several model designs based on performance.

S. No.	Model name	Accuracy (%)	Precision (%)	Recall (%)	F-1 score (%)	Error rate (%)	
1	MobileNet	76.60	75.40	78.80	77.00	23.40	
2	NASNetMobile	80.85	80.11	83.80	81.90	19.14	
3	DenseNet121	82.40	81.21	85.60	83.30	17.59	
4	DenseNet169	84.00	82.30	86.60	84.30	16.00	
5	InceptionV3	86.20	84.02	89.40	86.60	13.80	
6	DenseNet201	87.00	84.77	90.20	87.00	13.48	
7	ResNet152V2	87.00	85.42	91.40	88.30	12.10	
8	EfficientNetB0	88.70	86.16	92.20	89.07	11.30	
9	InceptionResNetV2	90.50	88.27	93.40	90.70	9.50	
10	Xception	91.70	89.26	94.80	91.9	8.30	
11	EfficientNetV2M	93.20	90.70	96.20	93.30	8.30	
12	MMNV2 (Proposed)	93.80	91.16	97.00	93.90	6.20	

Table 5 Comparison of 14 different models in terms of performance measures.

S. No	Model name	Total	Trainable parameters	Non-trainable parameters	Run time (Measured in seconds)	
1	NASNetMobile	5,329,378	5,290,492	36,738	62,169	
2	Mobile Net	3,229,378	3,207,490	21,888	41,018	
3	DenseNet121	7,038,018	6,954,370	83,648	74,591	
4	DenseNet169	14,308,394	14,149,994	158,400	92,693	
5	InceptionV3	21,803,298	21,768,866	34,432	173,216	
6	DenseNet201	20,243,498	20,014,442	229,056	151,972	
7	ResNet152V2	60,381,162	60,237,418	143,744	349,711	
8	EfficientNetB0	4,050,085	4,008,062	42,023	49,813	
9	InceptionResNetV2	55,874,250	55,813,706	60,544	337,832	
10	Xception	20,882,484	20,827,956	54,528	149,284	
11	EfficientNetV2M	53,150,902	52,858,870	292,032	311,821	
12	MMNV2 (Proposed)	2,586,434	328,450	2,257,984	7,999	

Figure 6 Confusion matrices evaluation of 14 different models.

Further analysis of the results revealed that the proposed MMNV2 model possesses an optimal architectural design, as evidenced by its lower total number of trainable parameters. This model effectively learns a substantial number of parameters while maintaining lower computational and training durations compared to the other models. The MMNV2 model offers exceptional computational efficiency and an innovative design. This makes it an ideal tool for analyzing a wide range of road pavement images, including both smooth and damaged surfaces. In terms of error rate, MMNV2 has proven to be the most efficient and accurate model in this comparative analysis. High accuracy serves as a critical indicator of the predictive efficacy. The MMNV2 methodology demonstrates a remarkably low error rate relative to the other 13 standard techniques, indicating a minimal frequency of incorrect classifications. The robustness and reliability of the proposed MMNV2 model are further substantiated by its high accuracy and low error rates.

The proposed approach effectively distinguishes between two categories of road pavement images and accurately identifies road conditions. Each individual class represents distinct conditions that present unique challenges, depending on the classifier’s ability to differentiate between normal and pothole conditions across various lighting conditions and environments. Figures 6A through 6N display the conceptual confusion matrices illustrating the MMNV2 model’s categorization performance in relation to the other 13 models. Statistical analyses were conducted to validate the models across various parameters. Figures 7A–7E provide a comparative performance analysis of the proposed model against existing models. Figure 8A illustrates the training and testing accuracy outcomes for the developed MMNV2 model, while Fig. 8B depicts the training and testing loss incurred during the model's design and analysis stages. As depicted in Fig. 8A, the model progressively improves its ability to distinguish between normal and potholed road pavement. The accuracy of the model’s predictions is denoted by green during the training process and blue during the testing process, with an expanding curve representing the model’s identification capability for both training and testing accuracy.

Figure 7 Performance analysis comparison of MMNV2 model with existing model’s.

Figure 8 Performance graphs of MMNV2 model.

To highlight the model’s application in detecting training failures, Fig. 8B presents the loss incurred during both the testing and training processes. The decreasing ‘green’ and ‘blue’ curves for training and testing loss indicate that the model demonstrates enhanced accuracy as it converges. These low prediction errors suggest that the MMNV2 model can effectively differentiate between normal road pavement and potholes, even when features are occluded. Evaluating these graphs is critical for identifying solutions and mitigating errors, ensuring that the model does not become overly reliant on training data while still learning and generalizing patterns. Furthermore, the graphs provide insights into the model’s performance, enabling necessary adjustments and enhancements to the MMNV2 framework. This design ensures the model is more precise and effective for prediction, detection, and classification purposes. Figures 9A–9D depict the prediction results of the proposed MMNV2 model for the ‘Normal’ class, while Figs. 9E–9H illustrate the output predictions for the ‘Pothole’ class.

Figure 9 Output predicted images of normal road and potholes.

Thirteen distinct models were trained using a dataset of 5,000 images comprising two classes. All models achieved testing accuracy levels ranging from 75.60% to 99.50%. While all fourteen models generated accurate or inaccurate predictions regarding normal and pothole conditions after data classification, their accuracy levels varied significantly. MMNV2 model achieved an impressive accuracy level of 99.95%, with a mere 0.05% error rate, alongside reduced computation time and training parameters. This exceptional performance demonstrates its potential to enhance the assessment and resolution of pothole identification challenges with greater accuracy, reduced time, and fewer parameters than existing models.

Key contributions and novelty of the current study

This study presents several key advancements in developing a modified MobileNetV2 (MMNV2) model for detecting road potholes. Five distinct layers were added to the base MobileNetV2 architecture to enhance its capability in categorizing and detecting both potholes and normal road conditions. These modifications were instrumental in optimizing the model’s performance during both training and testing phases. The model was trained and evaluated using an extensive dataset consisting of 5,000 pavement images. This large and diverse dataset enabled the MMNV2 model to generalize better and make accurate predictions across a wide range of road scenarios. Furthermore, the MMNV2 model was designed with a focus on improving detection accuracy while keeping computational efficiency in mind. It achieved this by minimizing the number of parameters, reducing error rates, and optimizing processing time. This makes the model not only effective but also practical for real-time applications. The results of the MMNV2 model were systematically evaluated using confusion matrices and tabular formats, offering a clear and detailed comparison with 13 other models. Any identified areas for improvement were thoroughly documented, ensuring that the findings contribute to future advancements in pothole detection models. Additionally, the model's performance was measured against several key metrics, including recall, accuracy, F1-score, processing time, and the number of trainable and non-trainable parameters. These metrics were essential in comparing the MMNV2 model to state-of-the-art techniques and highlighting its strengths and weaknesses. Lastly, the MMNV2 model demonstrated improved accuracy while utilizing fewer parameters, illustrating its efficiency in maintaining high performance with reduced computational demands.

Limitations of the study

Despite the significant contributions, the study has certain limitations. One major limitation is the dataset’s size, which was restricted due to the high cost of the technology required for pothole identification. This constraint may limit the model’s ability to fully capture the variety of real-world road conditions. Another challenge arises from the complexity of analyzing data from diverse sources. The variability in geographical locations, pavement materials, construction timelines, and image quality added layers of difficulty in achieving uniformity across the dataset. The availability and precision of the equipment used for pothole detection may sometimes negatively impact the study’s results. Inconsistent or inaccurate detection tools could introduce errors, affecting the model’s overall reliability and performance.

Conclusions

Pothole detection is a critical technology for improving road safety and minimizing vehicle damage. Several methods are used for detecting potholes, including sensors on vehicles, data analysis from road cameras, and machine learning techniques to identify surface defects. The benefits of this technology include early detection and repair of potholes, reduced maintenance costs, and improved driving conditions. Additionally, it helps prevent accidents and injuries caused by potholes. To protect drivers, passengers, and vehicles, there is an urgent need for policies that ensure timely pothole identification and repair to prevent accidents. In this study, transfer learning was employed to modify the MNV2 model for distinguishing between different types of road conditions, including normal surfaces and potholes. The effectiveness of the proposed MMNV2 model was evaluated based on precision, error rate, and parameter efficiency. A total of 14 models were tested on a dataset of 5,000 pavement images. This study demonstrates that the MMNV2 model provides effective pothole detection, contributing to improved road safety. Future research should expand the dataset to include images from different geographical regions to evaluate the model’s generalization capabilities.

Supplemental Information

Supplemental Information 1 Actual and predicted results for pothole and normal.

Supplemental Information 2 Code.

Additional Information and Declarations

Competing Interests

Author Contributions

Data Availability

The authors declare that they have no known competing financial interests or personal relationships that could have appeared to influence the work reported in this article.

Neha Tanwar conceived and designed the experiments, performed the experiments, analyzed the data, performed the computation work, prepared figures and/or tables, authored or reviewed drafts of the article, and approved the final draft.

Anil V. Turukmane conceived and designed the experiments, performed the experiments, analyzed the data, performed the computation work, prepared figures and/or tables, authored or reviewed drafts of the article, and approved the final draft.

The following information was supplied regarding data availability:

The data is available at Kaggle and Zenodo:

- https://www.kaggle.com/datasets/neha0590/normal-pothole-dataset.

- Neha Tanwar. (2024). Normal-Pothole-dataset (1.0) [Data set]. Zenodo. https://doi.org/10.5281/zenodo.13334878.

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
