# Peer review of "Modified MobileNetV2 transfer learning model to detect road potholes"

_PeerJ Computer Science, doi:10.7717/peerj-cs.2519_

## Round 0.1 · original submission · Major Revisions

Dear authors,

Thank you for submitting your article. Feedback from the reviewers is now available. It is not recommended that your article be published in its current format. However, we strongly recommend that you address the issues raised by the reviewers, especially those related to readability, experimental design and validity, and resubmit your paper after making the necessary changes. Before submitting the paper following should also be addressed:

1. Please write research gap and the motivation of the study. Evaluate how your study is different from others. Please highlight the originality and advantages of the proposed method. More recent literature should be examined.

2. All of the values for the parameters of the algorithms selected for comparison should be given.

3. Equations should be used with equation number. "F1-score, an essential metric for assessing the effectiveness of a classification model, is computed utilizing Equation (3)" should be corrected.


**Language Note:** The review process has identified that the English language must be improved. PeerJ can provide language editing services - please contact us at [email protected] for pricing (be sure to provide your manuscript number and title). Alternatively, you should make your own arrangements to improve the language quality and provide details in your response letter. – PeerJ Staff
Best wishes,

Reviewer 1 ·

Basic reporting

The paper has several issues with its writing quality. Some sections are unnecessarily repetitive, indicating a need for significant improvement in writing technique. It is recommended that the authors have the paper thoroughly proofread before resubmission to address these issues.

The manuscript contains many grammatical errors and typographical issues, such as incorrect capitalization. These errors detract from the professionalism and readability of the paper.

The abstract should be revised to clearly and concisely state the problem being addressed. Additionally, the term ‘colon’ in the abstract is unclear and should be clarified.

There is improper use of technical terminology, such as using ‘picture processing’ instead of the correct term ‘image processing.’

Figures:
- The sub-figures (a, b, etc.) should be clearly described in the figure captions to improve clarity. Severl figures have this problem
- Figure 3 is labeled as “Block Diagram of MMNV2 model,” but it does not represent a block diagram of the model. The figure is a mix of processes and model components and lacks systematic organization.
- There is no need to show all confusion matrices. The information can be presented in a better and more efficient way

Experimental design

I have found inconsistent description of the number of images that is actually used in the work. How many images are actually used in the training?

Validity of the findings

There are inconsistencies between the accuracy and other metrics reported in the text and those presented in the tables. These discrepancies need to be resolved to ensure the accuracy of the results.

Technically, the authors should discuss the potential drawbacks of adding the extra five layers, such as the impact on computational efficiency.

Additional comments

The authors should rewrite the entire paper to be more concise and systematic.
The paper should be thoroughly proofread before resubmission.

Cite this review as

Reviewer 2 ·

Basic reporting

- Fair use of English in the through though occassional typos.
- Most relevant papers related to the research topic cited.
- Paper's structure and logical arguments and workflow are resonably embraced in the relevant sections.
- Some raw data and pavement images are fine, perhaps more images are needed to improve the visibility of pavement deteriorations.
- I did not spot specific hypotheses.
- sufficient definitions of terms and theoretical backgrounded incoporated.

Experimental design

Additional experimental results can enhanced the readability and convencingly demonstrated the usefulness of its conclusions.

Validity of the findings

I have no means and reserach tools to verify the validity of the experimental results.

Additional comments

Nil

Cite this review as

Reviewer 3 ·

Basic reporting

The manuscript is poorly written.

Experimental design

The experimental design is not sound and not novel at all.

Validity of the findings

The contribution of this manuscript is limited.

Additional comments

The manuscript is poorly written, and the contribution of this manuscript is limited. A lot of similar work has been done in this area. Specific issues have been listed below:
1. In line 45, replace “picture processing” with “image processing”
2. In line 114, CNN belongs to Deep learning.
3. In line 132, it is “monitoring”
4. In line 153, the sentence is incomplete.
5. The introduction is not adequate. What is the research gap?
6. A lot of common sense was introduced in section 3.1. It is tedious in this manuscript. However, the advantages of the proposed model is not ideal and the details have not been explained well.
7. Why were 5 layers added into the proposed model? Have you compared the performance with different number of layers?

Cite this review as

---

## Round 0.2 · Minor Revisions

Dear authors,

Thank you for addressing the reviewers' concerns. Two of the three reviewers did not accept to review of the revised manuscript. The other reviewer thinks that just a single typo needs to be fixed. When submitting the revision, the following should also be addressed:

1. English grammar and writing style errors must be fixed. The paper still needs proofreading. Some paragraphs seem to be translated using tools without further clear corrections by the authors.
2. "...(CNNs) methods. [14]." should be corrected.
3. Blank characters should be correctly used. See line 60 "accidents[6]", line 758 "...1.As the..." and etc.
4. Organization of the paper is wrongly provided. There is not "Section 6".
5. All variables should be written in italic as in the equations. Ther definitions and possible boundaries should also be provided.
6. Values of the variables for the models should be provided.

Best wishes,

Reviewer 3 ·

Basic reporting

In figure 1, revise the wrong caption “pathole” to “pothole”.

Experimental design

no comment

Validity of the findings

no comment

Additional comments

no comment

Cite this review as

---

## Round 0.3 · Major Revisions

Dear authors,

Thank you for the revision. Although one reviewer thinks your article is acceptable, one reviewer still has major concerns and criticisms. We encourage you to perform the necessary additions and modifications specified by Reviewer 1 and to resubmit your article once you have updated it accordingly.

Best wishes,

Reviewer 1 ·

Basic reporting

After a careful review of the manuscript titled “Modified MobileNetV2 Transfer Learning Model to Detect Road Potholes,” I regret to inform you that the paper, in its current form, has not addressed the previous feedback sufficiently and does not meet the necessary standards for publication. The primary concern lies with the writing quality, which significantly affects the clarity, flow, and overall readability of the paper. As a result, the ideas presented are difficult to follow, and many sections are ambiguous or inaccurate. I strongly recommend the authors thoroughly revise the manuscript, paying close attention to improving the writing technique before resubmission. Below, I detail specific shortcomings:

1. Abstract:
• The abstract states, “for road pavement classification and detection,” but this is misleading. The correct terminology should be “pavement pothole classification and detection.”
2. Writing Clarity:
• The sentence, “Vehicle damage therefore weakens the pavement structure and destroys certain sections of the road surface,” is awkward and unclear. A clearer version could be: “Vehicle damage weakens the pavement structure and leads to the degradation of specific sections of the road surface.”
3. Terminology Usage:
• The term “boosting security” is incorrect in this context. The correct phrase should be “boosting safety,” as this is the relevant focus for road infrastructure.
4. Organization of Information:
• In the section where methods for pothole detection are described, the authors have presented the information in a numbered list. It would be more appropriate to present this information in paragraph form to improve flow and coherence.
5. Unclear Sentences:
• The sentence, “Stereo vision depends on methods constructed for deep-learning pothole identification,” is unclear and needs to be rewritten for clarity. The authors should specify the relationship between stereo vision and deep learning more clearly.
6. General Writing Quality:
• Many sentences throughout the manuscript are poorly constructed and exhibit improper English usage. The paper would greatly benefit from professional proofreading and editing services before resubmission.
7. Content Relevance:
• In Section 1.1, the discussion unexpectedly shifts to road cracks rather than focusing on pothole detection, which is the central theme of the paper. This detracts from the focus and should be revised for consistency.
8. Terminology:
• In Section 1.1, line 7, the word “splits” is used incorrectly. It seems the authors meant to refer to “cracks” in the road surface. This needs correction.
9. Writing Style:
• The writing style throughout the manuscript needs significant improvement. Many terms are used inaccurately, and certain phrases are awkward or inappropriate. A more careful, technical tone should be adopted, ensuring terms are used consistently and correctly.
10. Consistent Terminology:
• The term “craters” is used to describe potholes in certain parts of the paper. This is inconsistent. The manuscript should consistently use the term “potholes” for clarity and precision.
11. Inaccurate Terms:
• The manuscript states that “bumps” are an irregularity in pavements. However, this terminology is unclear. The authors should clarify whether they mean “uneven surfaces” or are referring to something else, as bumps do not necessarily indicate pavement irregularities.
12. Complex Sentences:
• The sentence, “This category includes devices made by Microsoft such as Kinect, cameras, and laser scanners, along with segmentation, 3-dimensional point cloud modeling, normal 2-dimensional image processing, and the development of a cutting-edge computer vision algorithm known as Sota,” is convoluted and unclear. This sentence should be broken down and rewritten for clarity, specifying each component in a coherent manner.
13. Data Inconsistencies:
• The manuscript inconsistently mentions the total number of images used in the study. At one point, it states that there are 5000 images in total, but later, it says that 5000 images were used for training, which constitutes 80% of the total. This contradiction should be clarified, as it affects the reliability of the experimental results.

Experimental design

Please see the comments above.

Validity of the findings

Please see the comments above.

Additional comments

Due to the issues mentioned above, particularly the poor writing quality, numerous inaccuracies, and inconsistent terminology, the paper is currently not suitable for publication. I recommend that the authors substantially revise the writing, ensuring the technical content is communicated clearly and accurately. A professional proofreading service is highly recommended before resubmission to improve readability and ensure the paper meets the necessary standards.

Cite this review as

Reviewer 3 ·

Basic reporting

No further comments.

Experimental design

No further comments.

Validity of the findings

No further comments.

Additional comments

No further comments.

Cite this review as

---

## Round 0.4 · accepted · Accept

Dear Authors,

Thank you for the revision. One of the previous reviewers did not respond to the invitation for revised paper and other reviewer thinks that the paper can be accepted in this form. I also assessed the revision and I am happy with the current version. The paper now seems acceptable for publication.

Best wishes,

Reviewer 2 ·

Basic reporting

1) The revised version is much clearer with polished English.
2) Literature references provide most relevant material, though not exhaustive.
3) Text body structure and logic flow are smooth and more readable without deep domain specific knowledge needed.
4) The results are comprehensive presented and discussion is insightful.
5) Formal results are clear with relevant terms and theorem with proof.

Experimental design

Interesting results included.

Validity of the findings

Fair internal and external validity of the obtained results.

Additional comments

This version is now in publishable form.

Cite this review as